# Tick-Borne Diseases of Humans and Animals in West Africa

**DOI:** 10.3390/pathogens12111276

**Published:** 2023-10-24

**Authors:** Adama Zan Diarra, Patrick Kelly, Bernard Davoust, Philippe Parola

**Affiliations:** 1IHU-Méditerranée Infection, 13005 Marseille, France; adamazandiarra@gmail.com (A.Z.D.); bernard.davoust@gmail.com (B.D.); 2Aix Marseille Univ, IRD, AP-HM, SSA, VITROME, 13005 Marseille, France; 3Ross University School of Veterinary Medicine, Basseterre P.O. Box 334, Saint Kitts and Nevis; pkelly@rossvet.edu.kn; 4Aix Marseille Univ, IRD, AP-HM, MEPHI, 13005 Marseille, France

**Keywords:** ticks, tick-borne diseases, *Rickettsia*, *Borrelia*, West Africa

## Abstract

Ticks are a significant group of arthropod vectors that transmit a large variety of pathogens responsible for human and animal diseases worldwide. Ticks are the second biggest transmitters of vector-borne diseases, behind mosquitoes. However, in West Africa, there is often only limited knowledge of tick-borne diseases. With the scarcity of appropriate diagnostic services, the prevalence of tick-borne diseases is generally underestimated in humans. In this review, we provide an update on tick-borne pathogens reported in people, animals and ticks in West Africa by microscopic, immunological and molecular methods. A systematic search was conducted in PubMed and Google Scholar. The selection criteria included all studies conducted in West Africa reporting the presence of *Rickettsia*, *Borrelia*, *Anaplasma*, *Ehrlichia*, *Bartonella*, *Coxiella burnetii*, *Theileria*, *Babesia*, *Hepatozoon* and Crimean–Congo haemorrhagic fever viruses in humans, animals or ticks. Our intention is to raise awareness of tick-borne diseases amongst human and animal health workers in West Africa, and also physicians working with tourists who have travelled to the region.

## 1. Introduction

Ticks are obligate blood-feeding parasites belonging to the phylum Arthropoda, the class Arachnida, and the order Acari [1]. They are divided into two main families: Ixodidae (hard ticks), comprising over 700 species worldwide, and Argasidae (soft ticks), comprising roughly 200 species [2]. Currently, in Africa, domestic animals can be infested with up to ten genera of ticks seven Ixodidae and three Argasidae [1]. A recent study reported that several species and subspecies of hard ticks, notably of the genera *Hyalomma*, *Rhipicephalus*, *Ixodes*, and *Amblyomma*, infest animals in different parts of the continent [3]. While ticks prefer animal hosts, many species will also feed on humans [4]. Globally, after mosquitoes, ticks are the second biggest group of vectors and reservoirs of animal and human viral, bacterial and protozoal pathogens [5]. More than 80% of the world’s cattle population is at risk of contracting tick-borne diseases (TBDs) [6] which are, particularly in sub-Saharan Africa, a major threat to the health, welfare and productivity of livestock. The impact ticks have on their hosts relates to their blood-feeding which can lead to both direct pathologies (e.g., anaemia, skin infections) and indirect pathologies (e.g., pathogen transmission, immunosuppression) [7]. Tick-borne diseases are strongly influenced by many factors, including host distribution, tick abundance and seasonality, pathogen virulence, and climate (temperature, precipitation, humidity, and vegetation cover), all of which contribute to the emergence and re-emergence of tick-borne diseases. Climate change may impact the incidence of tick-borne diseases by increasing tick populations, the rate of contact between livestock and ticks, and the rate of contact between livestock and wildlife [8]. In Europe, studies have been carried out on the impact of climate change on tick-borne diseases. It has been reported that climate change is responsible for the extension of the range of *Ixodes ricinus* in the north and at higher altitudes, leading to an increase in the prevalence of tick-borne encephalitis. Climate change is also partly responsible for changes in the distribution of *D. reticulatus* [9]. Over the last 20 years, the incidence of tick-borne diseases (Lyme disease, tick-borne encephalitis, and Crimean–Congo haemorrhagic fever) has increased in both Europe and America [10]. Although this increase could be partly caused by climate change, other factors could also contribute, as tick-borne disease systems are quite complex [10]. In Africa, diseases transmitted by ticks to humans are poorly studied, making it difficult to measure the impact of climate change on these diseases. However, there has been an increase in the incidence of tick-borne relapsing fever, one of the best-studied tick-borne bacterial diseases, in Senegal since the 1970s, and its range has extended over 350 km to north-west Morocco due to increased drought conditions [10].

The term “West Africa” is used to describe the land spanning the entire western part of sub-Saharan Africa. Roughly, this includes the coastal countries north of the Gulf of Guinea to the Senegal River, the countries covered by the Niger River Basin, and the countries of the Sahel hinterland [11]. This 6,140,000 km^2^ zone represents one-fifth of the area of the African continent and includes 16 countries, namely Benin, Burkina Faso, Cape Verde, Côte d’Ivoire, Gambia, Ghana, Guinea, Guinea Bissau, Liberia, Mali, Mauritania, Niger, Nigeria, Senegal, Sierra Leone, and Togo [11] (Figure 1).

In this review, we use a “One Health” approach to describe human and animal TBDs in West Africa. The One Health concept encourages cooperation between professionals working on animal, environmental, and human health to develop integrated solutions for complex problems that impact on the health of animals, humans, and the planet [12]. To carry out this review, we searched the PubMed and Google Scholar databases using the keywords: ticks, tick-borne diseases, humans, animal, West Africa, *Rickettsia*, *Borrelia*, *Anaplasma*, *Ehrlichia*, *Bartonella*, *Coxiella*, *Theileria*, *Babesia*, *Hepatozoon*, and Crimean–Congo haemorrhagic fever virus. Papers that were published between 1970 and 2020, including molecular biology and serology studies, were selected for review and included in this review if they were written in English or with an English abstract.

## 2. Bacterial Diseases

### 2.1. Tick-Borne Spotted Fever Group Rickettsioses

Bacteria of the genus *Rickettsia* are obligate intracellular organisms that comprise 31 species of which 17, to date, are considered human and/or animal pathogens [13]. The spotted fever group (SFG) of *Rickettsia* comprises around 30 species that, with the exception of *Rickettsia felis* which is transmitted by fleas and possibly mosquitoes [14,15,16], are transmitted mainly by ticks and can cause tick-borne spotted fever group rickettsioses in animals and/or people. The typhus group comprises two species, *Rickettsia typhi* and *Rickettsia prowazekii*, which are most commonly transmitted by fleas and human body lice, respectively, but have occasionally been associated with ticks [13]. In humans, tick-borne spotted fever group (SFG) rickettsioses manifest mainly as fever, headache, muscle pain, and a rash which may be maculopapular but also vesicular or purpuric in severe cases. There might also be an inoculation eschar at the site where the tick was attached and an inconsistent regional lymphadenopathy [13]. Laboratory investigations may show thrombocytopaenia, hyponatraemia, elevated transaminases, and hyperbilirubinaemia, and the diagnosis can be made by serological (immunofluorescence) or molecular (quantitative PCR or standard PCR, particularly efficient on eschar swabbing samples) methods [13,17].

Five pathogenic SFG rickettsia have been reported in West Africa: *Rickettsia africae*, the agent of African tick-bite fever (ATBF), *R. conorii conorii*, the agent of Mediterranean spotted fever (MSF), *R. sibirica mongolitimonae*, the agent of lymphangitis-associated rickettsioses, and *R. aeschlimannii* and *R. massiliae*, which are agents of emerging rickettsioses in humans [13] (Figure 1).

*Rickettsia africae* is transmitted by ticks of the genus *Amblyomma*, in particular *Amblyomma*
*variegatum* in West Africa (Figure 2). The infection rate of *R. africae* in these ticks, which feed very readily on people, is often very high, up to 100% [9]. After malaria, ATBF is the most commonly documented aetiology of fever in travellers returning from sub-Saharan Africa [18]. More than 350 travel-associated cases of ATBF have been reported in travellers from EU member countries, North America, Australia, Argentina, and Japan. Most of these travellers were infected in South Africa. In West Africa, cases of ATBF have only been reported in travellers who had stayed in Gambia [18]. As is often the case in the rest of Africa, in West Africa ATBF is often poorly recognised by health workers, perhaps due to lack of knowledge of the disease and diagnostic laboratories, but also because the disease is most often benign, and eschars and mild rashes are more difficult to detect in black skin [13]. However, exposure to *R. africae* is very common in West Africa, with a prevalence of reactive antibodies reported to be 19.6% among blood donors and patients from Mauritania [19], between 1.1% and 25.4% in Guinea [20], and between 20.6% and 45.6% in malaria-negative patients with a recent fever in the villages of Dielmo and Ndiop in Senegal, respectively [21]. In the only serological study of animals from West Africa, the prevalence of antibodies against *R. africae* in farm animals ranged from 0.6% to 18.8% in Guinea [20]. It is important to note, however, that serological diagnostic methods are limited by cross-reactivity with other species of SFG *Rickettsia* and thus infections with other species may have contributed to the results.

In tick studies, DNA from *R. africae* has been found in *Am.*
*variegatum* from Senegal, Benin, Mali, Côte d’Ivoire, Niger, Liberia, Guinea, Nigeria, and Burkina Faso [22,23,24,25,26,27,28,29,30,31,32,33], and in *Amblyomma compressum* from Liberia [26] (Table 1). Its DNA has also been found in a variety of other tick species including *Hyalomma rufipes* from Mali, Côte d’Ivoire, Guinea and Senegal [23,24,26,31], *Hyalomma truncatum* from Mali and Côte d’Ivoire [23,24], *Hyalomma impeltatum* from Nigeria [28], *Hyalomma impressum* from Côte d’Ivoire [24], *Haemaphysalis paraleachi* from Guinea [26], *Rhipicephalus* (*Boophilus*) *annulatus* from Senegal, Guinea, and Nigeria [21,26,30], *Rhipicephalus evertsi evertsi* from Mali, Senegal, and Nigeria [21,23,30], *Rhipicephalus* (*Boophilus*) *decoloratus* from Guinea and Nigeria, *Rhipicephalus* (*Boophilus*) *geigyi*, *Rhipicephalus sanguineus* s.l., and *Rhipicephalus* (*Boophilus*) *microplus* from Liberia, Nigeria, and Côte d’Ivoire, respectively [24,26,27,30], and recently in *Rh.* (*B*.) *microplus* from cattle from Mali [23] (Table 1). To date, only *Amblyomma hebraeum* and *Am. variegatum* have been shown to be vectors and reservoirs of *R. africae* [32]. The prevalence of *R. africae* in *Am. variegatum* is generally between 90% and 100%, due to the high rate of transovarian transmission and filial infection. The presence of *R. africae* DNA in other genera, often with varying prevalence (0.4% to 93%), does not prove that these ticks are also vectors, as they may have contained organisms in their digestive tract from blood meals taken from rickettsemic hosts, or from probable transovarial transmission [13]. Further studies are required to test the vector competence of these other tick species for *R. africae*.

*Rickettsia conorii conorii*, the agent of MSF mainly transmitted by *Rh. sanguineus* s.l. [13], is one of the first disease agents which has been shown to be transmitted by arthropods. The disease is endemic in the Mediterranean region, where the majority of cases are encountered in the warmest months [75]. MSF is characterised by high fever, flu-like symptoms, a tache noir or ‘black spot’ representing an area of skin necrosis at the site of tick attachment, and a maculopapular rash. Neurological disorders and coagulopathies with multiple organ failure can occur in the severe forms of the disease [76]. In West Africa, *R. conorii conorii* has only been reported once, in a molecular study of a patient in Senegal who was negative for malaria but had a history of fever [21]. Bearing in mind the lack of specificity of serological tests (above), antibodies against *R. conorii* in indirect immunofluorescent antibody assays (IFAs) have been found in 13.5% of blood donors from Mauritania [19], and 38.2%, 31.5%, and 27% of apparently healthy adults (blood donors or volunteers) in Burkina Faso, Côte d’Ivoire, and Mali, respectively [77]. In the latter study, antibodies were detected using Western blots in 47.8%, 34.8%, and 30% of the patients. In ticks, DNA of *R. conorii conorii* was detected in 2/100 *Rh. Sanguineus* s.l. collected from rodents in Nigeria [70] and in 1/2360 *Rh. E. evertsi* collected in Senegal [21] (Table 1).

*Rickettsia aeschlimannii* was first isolated from *Hyalomma marginatum*, but several other tick species, particularly other *Hyalomma* spp., have now been reported as potential vectors [13,76]. The pathogenicity of this organism was demonstrated for the first time in France in a patient who travelled to Morocco in August 2000 and presented with a vesicular lesion of the ankle, a black spot with a fever of 39.5 °C, and a generalised maculopapular rash [78]. A second case was reported in a South African man who had been bitten by a *Rhipicephalus appendiculatus* during a hunt and was symptomless as he self-prescribed doxycycline immediately after removing the tick [76]. In West Africa, *R. aeschlimannii* has not been detected in humans and domestic animals. However, the DNA of *R. aeschlimannii* has been found in *Hy. truncatum*, *Hy. rufipes*, and *Rh. sanguineus* s.l. from Mali [23], *Hy. rufipes* collected in Côte d’Ivoire, Senegal, Burkina Faso, and Nigeria [21,24,31,33,52], *Hy. truncatum* collected in Côte d’Ivoire, Senegal, and Burkina Faso [21,24,33], *Hy. impeltatum* collected in Senegal and Nigeria [31,52], *Rhipicephalus evertsi evertsi* collected in Senegal [21,31], *Am. variegatum*, *Rh. microplus*, and *Hy. rufipes* collected in Benin [34], and *Hy. rufipes*, a *Hy. impeltatum*, *Hy. truncatum*, and *Hy. dromedarii* from Nigeria [49] (Table 1).

*Rickettsia massiliae* is transmitted by *Rh. sanguineus* s.l. and has been associated with fever, a palpable purpuric rash, and an eschar in humans [13]. Although human infections have been reported in South America and Europe, in West Africa the organism has only been detected in animal studies with the DNA of *R. massiliae* having been found in 3.5% of cattle studied in Nigeria [55]. In ticks, *R. massiliae* has been detected in *Rhipicephalus* sp. and *Rh. senegalensis* from Côte d’Ivoire [24,79], in *Rh. sanguineus* and *Rh. turanicus* from Nigeria [27,74], and in *Rh. guilhoni* and *Rh. senegalensis* from Senegal and Liberia, respectively [21,26] (Table 1).

Finally, *R. sibirica mongolitimonae*, which has been associated mainly with ticks of the genus *Hyalomma*, has been reported in humans in Southern Europe but also in travellers returning from North Africa (Algeria and Egypt) [13]. The clinical signs in patients include fever, maculopapular rash, one or more eschars, and regional lymphadenitis and lymphangitis that is characteristic of the disease and leads to it being named “rickettsiosis associated with lymphangitis” [76]. No cases of *R. mongolitimonae* rickettsiosis have been reported in West Africa in humans or animals. However, the DNA of *R. mongolitimonae* was detected in *Hy. truncatum* ticks collected from cattle in Senegal and Mali [21,23] (Table 1).

Two undescribed *Rickettsia* species have been reported from West Africa, including “*Candidatus Rickettsia liberiensis*”, genetically related *to R. raoultii* detected in *Ixodes muniensis* ticks collected from dogs in Liberia [26], and a *Rickettsia* species belonging to the *Rickettsia rickettsii* group in the tick *Rh. evertsi* in Nigeria [30]. Recently, the DNA of *Rickettsia* spp. was detected in *Am. latum*, *Am. variegatum*, *Ixodes aulacodi*, *Rh. sanguineus*, *Hy. rufipes*, *Hy. truncatum*, *Rh. annulatus*, and *Rh. microplus* collected from snakes, four-toed hedgehogs in Benin, cattle in Benin and Togo [34,36], and in *Am. variegatum*, *Rh. sanguineus* s.l., *Rhipicephalus* sp., *Hy. truncatun*, *Hy. rufipes*, *Rh*. *evertsi*, and *Rhipicephalus* (*Boophilus*) sp. in Ghana [35]. The different *Rickettsia* species detected in West Africa in humans, animals, and ticks are shown on the map in Figure 2.

### 2.2. Tick-Borne Relapsing Fevers

The tick-borne relapsing fevers (TBRFs) are acute febrile illnesses characterised by multiple recurrences of fever, headache, myalgia, and arthralgia. They are caused by spirochetes of the genus *Borrelia* (*Borrelia crocidurae*, *Borrelia duttonii*, *B. recurrentis*, and *Borrelia hispanica*), which are endemic in subtropical regions around the world [80]. Historically, TBRFs were considered to be transmitted by soft ticks of the genus *Ornithodoros* [81]. However, in 2011, the paradigm of TBRF transmission by soft ticks changed with the dis-covery in Russia in 1995 of *Borrelia miyamotoi*, which was reported to cause TBRFs and be transmitted by hard ticks (*Ixodes* genus); this was subsequently confirmed in Europe, Japan, and the USA [81]. They are a major cause of febrile illness in several regions of Africa [60], and in West Africa TBRF is caused by *B. crocidurae* transmitted by *Ornithodoros sonrai* [60]. There are also *Borrelia* that cause disease in animals—mainly *Borrelia theileri* causing bovine borreliosis, and other *Borrelia* of unknown pathogenicity which have also been detected in humans and ticks [82]. Tick-borne relapsing fever can be diagnosed by microscopy (microscopic detection of spirochetes in blood smears), culture, and animal inoculation, but the best and most widely available tools are molecular methods (PCR or qPCR in blood samples) that have been shown to be more sensitive and specific than blood smears during the acute febrile phase [17,81,83].

In West Africa, TBRF has mainly been studied in Senegal [84] where the presence of *B. crocidurae* has varied widely, depending on the study, date, and location. Between 1989 and 1990, there was only a low prevalence of 0.9% (12/1340) in children under the age of 14 with acute fever presenting at the Keur Moussa dispensary [85], while between 1990 and 2003 the average incidence of TBRF was 11 per 100 person-years [86]. The DNA of *B. crocidurae* was reported in 5.1% of patients with a fever in Dielmo in 1996 [87], 0.42% of suspected malaria patients examined in Dakar between October 1999 and October 2003 [88], 15% and 0.3% of patients who tested negative for malaria in 2011 in Dielmo and Ndiop, respectively [83], and 9.5% of febrile patients from Dielmo and Ndiop between February 2011 and January 2012 [89]. Further, TBRF has been reported in patients returning from Senegal who developed a fever [90]. Recently *B. crocidurae* DNA was found in 11.7% and 7.22% of febrile patients in Senegal [57,91]. In Togo, the DNA of *B. crocidurae* and B. *duttonii* was detected in 8.8% and 1.2% of febrile patients, respectively [92], while *B. crocidurae* was found in 3.4% of febrile patients who tested negative for malaria in Mali [93]. *Borrelia crocudirae* has also been detected by a variety of methods in asymptomatic small mammals in Senegal, Mali, and Mauritania [57,58,62,94], and by demonstrating *B. crocidurae* in blood smears in 17.6% of 740 rodents and 7.3% of 55 musk shrews in Senegal [95]. In Mali, 11.3% (82/726) of animals (rodents and shrews) were positive for antibodies against relapsing fever spirochetes, while 2.20% (16/724) were positive for spirochetes in blood smears [59]. In ticks, *B. crocidurae* has been reported in *O. sonrai* from Mauritania, Senegal, Mali, and Gambia [56,57,58,59,60] (Table 1).

The agent of bovine borreliosis, *B. theileri*, has been reported in Africa, Australia, and North and South America in cattle, goats, and sheep. In cattle, infections usually manifest as fever and anaemia [66]. In West Africa, the DNA of *B. theileri* has been detected in *Rh.* (*B*.) *geigyi* collected from cattle in Mali [66] (Table 1).

Other *Borrelia* species of unknown pathogenicity for humans and animals have been detected in ticks and cattle in West Africa. *Borrelia* spp. were detected in domestic animals in Ghana [96], in ticks collected from vegetation in Nigeria [30], and in *Am. variegatum* and *Hy. truncatum* collected in Mali [23] (Table 1). Three potential new *Borrelia* species (*Candidatus* Borrelia Africana, *Candidatus* Borrelia ivorensis, and *Candidatus* Borrelia kalaharica) have been detected in West Africa, the first two of which were detected in *Am. variegatum* from Côte d’Ivoire [24] and the other in *Ornithodoros savignyi* and livestock from Nigeria [97,98] (Table 1). The different *Borrelia* species detected in West Africa in humans, animals, and ticks are shown on the map in Figure 3.

### 2.3. Anaplasmosis

Anaplasmosis is a disease of humans and animals that is caused by *Anaplasma* species, all obligate intracellular gram-negative bacteria, that are mainly transmitted by ticks (Figure 4) [99,100]. In humans, *A. phagocytophilum* can be found in circulating neutrophils and is the agent of human granulocytic anaplasmosis (HGA), manifesting as lethargy, inappetence, weight loss, musculoskeletal pain, respiratory insufficiency, and severe gastrointestinal bleeding [99,100]. Thrombocytopaenia and liver enzyme alterations are the most common laboratory abnormality in HGA [99]. In animals, *A. phagocytophilum* also infects neutrophils and is the agent of tick-borne fever in sheep, pasture fever in cattle, canine granulocytic anaplasmosis, and equine granulocytic anaplasmosis [101]. Signs common to all species include fever, depression, and inappetence, with sheep and cattle also developing leukopaenia and severe secondary infections, horses developing icterus, ventral oedema, and petechiation [101], and dogs developing gastrointestinal and respiratory signs [102]. Diagnosis is based on clinical signs and symptoms, including thrombocytopaenia including leukopaenia and elevated transaminases, by microscopic identification of morulae in neutrophils on a blood smear, or in the buffy coat by serological and molecular methods [17,103]. In West Africa, there have been no reports of HGA, but the DNA of *A. phagocytophilum* has been detected in one healthy dog from Nigeria [104] and in two febrile sheep from Senegal [105]. *Anaplasma phagocytophilum* is mainly transmitted by ticks of the genus *Ixodes*, the tick species of which vary according to geographical area [106]. For example, *I. pacificus* and *I. scapularis* in the United States, *I. ricinus*, *I. trianguliceps*, *I. hexagonus*, and *I. ventalloi* in Europe, and *I. persulcatus* in Asia and Russia [106]. In addition, DNA of *A. phagocytophilum* has been detected in other tick species, but their vectorial competence and their role in the epidemiological cycle of this bacterium are still not clear [106]. There have been no reported studies on *A. phagocytophilum* in ticks in West Africa, despite numerous studies being carried out in this area.

*Anaplasma marginale* is found in the erythrocytes of ruminants and is the agent of bovine anaplasmosis. It is mainly transmitted by ticks belonging to the *Rhipicephalus* and *Dermacentor* genera [106] and has a wide distribution in tropical and subtropical areas. The distribution of the organism around the world is spreading rapidly, probably due to the transport of cattle from endemic to non-endemic areas and global warming, which favours tick survival [106]. In West Africa, *A. marginale* has been detected in cattle in various areas of Senegal [107,108,109] by microscopy of blood smears, a technique which has poor sensitivity and specificity. Similarly, *A. marginale* antigens have been detected in ELISA studies of cattle from Gambia [110]. The DNA of *A. marginale* has been reported in cattle from Nigeria [55,111,112,113,114], Senegal [68], Ghana [112], Côte d’Ivoire [115], Burkina Faso, and Benin [42,44,112]. In ticks from West Africa, the DNA of *A. marginale* has been reported in *Rh.* (*B.*) *decoloratus* collected in Nigeria and Burkina Faso [28,44], in *Rh.*(*B*.) *microplus* from Côte d’Ivoire, Mali, Benin, and Guinea [23,24,37,65], in *Rh. geigyi* from Guinea [65], and in *Am. variegatum* from Benin [39] (Table 1).

*Anaplasma centrale* is responsible for clinically benign bovine anaplasmosis and is principally transmitted by the African tick *Rh. simus* [106]. As the organism confers protective immunity against *A. marginale* and is less pathogenic, it is used as a live vaccine against *A. marginale* infections in cattle in many countries [106]. In animals from West Africa, the DNA of *A. centrale* has been reported in cattle in Nigeria [55,112], Senegal [68], Benin [112], and Burkina Faso [42,112]. In ticks, the DNA of *A. centrale* was found in *Am. variegatum* from Côte d’Ivoire and *Rh.* (*B*.) *annulatus* and *Hy. impeltatum* from Nigeria [24,30] (Table 1).

*Anaplasma ovis*, responsible for anaplasmosis in sheep, goats, and wild ruminants in tropical and subtropical regions, is an obligate intra-erythrocytic bacterium transmitted by *Rhipicephalus* spp. [106]. *Anaplasma ovis* has been reported in Africa, Asia, Europe, and the United States [106]. In West Africa, it has been detected by microscopy in the blood of small ruminants in Senegal [105,107,108,109]. Its DNA has been detected in dogs from Nigeria [104], and in sheep from Senegal and Niger [68,116]. Currently, *A. ovis* has not been reported in ticks collected in West Africa, despite numerous studies carried out in this area.

*Anaplasma platys* is found in platelets in dogs worldwide and is the agent of infectious canine cyclic thrombocytopenia (ICCT). It is mainly transmitted by *Rh. sanguineus* s.l. but can be detected in *D. auratus*, *I. persulcatus*, and *Ha. longicornis*, although these ticks are not known to be involved in its transmission [106]. While *A. platys* is mainly a canine pathogen, it can occasionally be found in other animals such as cats, foxes, Bactrian camels, deer, sika deer, cattle, and humans [106]. In West Africa, the DNA of *A. platys* has been detected in dogs, camels, and cattle from Nigeria [55,113,117,118,119], dogs from the Maio Island in the Cape Verde archipelago [120,121], dogs from Ghana and Côte d’Ivoire [71,73,118], and dogs, cattle, goats, and sheep from Senegal [68,105]. In ticks, the DNA of *A. platys* has been identified in *Rh. sanguineus* s.l. collected from dogs in Côte d’Ivoire [71], in *Rh. microplus* collected from cattle in Guinea [65], and in *Hy. truncatum* collected from cattle in Nigeria [55] (Table 1).

Other unidentified species of *Anaplasma* have been reported in ticks, including ‘*Candidatus* Anaplasma ivorensis’, which is similar to *A. phagocytophilum* and has been identified in *Rh.* (*B.*) *microplus* collected from cattle in Mali [23], and *Am. variegatum* in Côte d’Ivoire [24]. In O. sonrai collected from Senegal, the DNA of *Anaplasma* sp. have been detected [58] (Table 1). In Senegal, the DNA of a new potential species of *Anaplasma* named *Candidatus* Anaplasma africae, *Candidatus* (Ca) Anaplasma turritanum, *Candidatus* (Ca) Anaplasma cinensis, *Candidatus* (Ca) Anaplasma africanum, and *Candidatus* (Ca) Anaplasma boleense were detected in cattle, sheep, and goats [68,122], *Anaplasma* spp. (Badiuoré Ziguinchor) in sheep in Senegal and dogs and cattle in Nigeria [38,55,105], and *Candidatus* Anaplasma camelii in the blood of camels from Nigeria [49].

Finally, *Anaplasma bovis* is mainly found in the monocytes of cattle and buffalo, but also in many other domestic and wild animals [106]. Several species of ticks are suspected to be vectors of this bacterium in Africa, including *Hyalomma* sp., *Am. variegatum*, *Rh. appendiculatus*, *Rh. sanguineus* s.l., and *Haemaphysalis* spp. [106]. To date, *A. bovis* has not been reported in West Africa.

The different *Anaplasma* species detected in West Africa in humans, animals, and ticks are shown on the map in Figure 4.

### 2.4. Ehrlichiosis

Ehrlichioses are diseases of animals and humans caused by tick-borne obligate intracellular gram-negative bacteria of the genus *Ehrlichia*. The genus *Ehrlichia* comprises five species [123], four of which have been reported in West Africa, namely *Ehrlichia canis*, *E. ewingii*, *E. ruminantium*, and *E. chaffeensis*, along with unidentified *Ehrlichia* species [82] (Figure 5).

*Ehrlichia chaffeensis* is found in circulating monocytes and is responsible for human monocytic ehrlichiosis (HME) that can have signs of fever, myalgia, headache, nausea, vomiting, acute renal failure, leukopaenia, thrombocytopaenia, and increased liver enzyme activity. The disease is regarded as an emerging zoonosis in the United States [124], with cases also reported in Europe and West Africa [125]. A person with HME has been described in Mali [126]. Antibodies against *E. chaffeensis* have been found in an asymptomatic blood donor in Burkina Faso, although serology assay cross reactions cannot be ruled out [127]. In the United States, *E. chaffeensis* is transmitted by *Am. americanum*, but DNA of the organism has been found in *Hy. impeltatum* in Nigeria [30] (Table 1). This study also identified three genotypes of *Ehrlichia* in *Rh. e. evertsi* that were phylogenetically close to *E. chaffeensis*.

*Ehrlichia ewingii* mainly infects granulocytes and is the aetiologic agent of granulocytic ehrlichiosis in humans and dogs in the United States where it is transmitted by *Am. americanum* [128]. In West Africa, no human or canine cases of granulocytic ehrlichiosis have been reported to date, but the DNA of *E. ewingii* has been detected in *Rh.* (*B.*) *annulatus* collected in Nigeria. Also, two other *Ehrlichia* genotypes that are closely related to *E. ewingii* were found in *Am. variegatum* and *Hy. impeltatum* [30] (Table 1).

*Ehrlichia ruminantium*, formerly *Cowdria ruminantium*, is the causative agent of heartwater in wild and domestic ruminants across Africa, where the main vectors are ticks of the genus *Amblyomma* [129]. Heartwater is a notifiable disease according to the World Organization for Animal Health (OIE) and a serious economic problem for pastoralists in sub-Saharan Africa. The disease occurs in most of sub-Saharan Africa, except the very dry south-west, and is present on the islands of Madagascar, Mauritius, Réunion, Grande Comore, and São Tomé. In the new world, *E. ruminantium* is present on the islands of Guadeloupe, Antigua, and Marie-Galante in the West Indies [129]. In West Africa, antibodies against *E. ruminantium* have been detected in small ruminants in Gambia [130], cattle in Côte d’Ivoire [131], and domestic ruminants in Ghana [132,133]. The DNA of *E. ruminantium* has been detected in lambs in Gambia [134], sheep in Senegal [105], cattle from Burkina Faso, Nigeria, Ghana, and Benin [40,55,112,135], and dogs in Nigeria [38]. The DNA of *E. ruminantium* has been found in *Am. variegatum* collected in Mali, Côte d’Ivoire, Burkina Faso, Gambia, Nigeria, and Benin [23,24,38,39,40,41,42]. Although it has also been detected in other species of ticks, including *Rh.* (*B.*) *microplus* collected on cattle in Mali, Côte d’Ivoire, and Burkina Faso [23,24,67], and *Hy. truncatum*, *Hy. rufipes*, and *Rh. e. evertsi* from Mali [23] (Table 1), the possible role, if any, that these ticks play in the epidemiology of *E. ruminantium* is undetermined.

*Ehrlichia canis* is transmitted by *Rh. sanguineus* s.l. and infects dogs worldwide, causing canine monocytic ehrlichiosis. Many infections are subclinical but there can be acute signs of fever, loss of appetite, lethargy, pallor, lymphadenomegaly, splenomegaly, and petechiation. In some chronically infected dogs, there is marked pancytopaenia with anaemia and secondary infections [123]. Antibodies against *E. canis* have only been reported in West Africa from dogs in Côte d’Ivoire [136,137]. The DNA of *E. canis* has been found in dogs from Nigeria [38,55,117,118,119,138,139,140,141], Senegal [68,142], Ghana [73], Cape Verde [120,121], and Côte d’Ivoire [143]. In ticks, the DNA of *E. canis* was detected in *Rh. sanguineus* s.l. collected from watchdogs in Côte d’Ivoire [72] and *Rh. eversti eversti* collected from sheep in Senegal [68] (Table 1).

The DNA of potentially new species of *Ehrlichia*, provisionally named “*Candidatus* Ehrlichia rustica” and “*Candidatus* Ehrlichia urmitei”, have been detected in *Am. variegatum*, *Hy. truncatum*, and *Rh.* (*B.*) *microplus* collected in Côte d’Ivoire and Mali [23,24]. *Candidatus* Neoehrlichia mikurensis has been identified in *Rh. sanguineus* s.l. in Nigeria [141], and *Ehrlichia* sp. (Omatjenne), which is phylogenetically close to *Ehrlichia ruminantium*, has been found *in Hy. truncatum* collected from cattle in Nigeria [55]. Two genotypes of *Ehrlichia* sp. (Erm58 and Eht224) from the *E. canis* group have been identified in *Rh. muhsamae* and *Hy. truncatum* collected in Mali and Niger, respectively [55] (Table 1). Recently, a potentially new species of *Ehrlichia*, *Candidatus* Ehrlichia senegalensis, and *Ehrlichia* sp., has been detected in rodents and *O. sonrai*, respectively, collected in Senegal [58,62]. The different *Ehrlichia* species detected in West Africa in humans, animals, and ticks are shown on the map in Figure 5.

### 2.5. Bartonellosis

The bartonelloses are caused by *Bartonella* species, which are fastidious haemotropic gram-negative organisms that can be transmitted by arthropods to people and a wide range of domestic and wild animals which can act as reservoir hosts [96]. At least 20 species of *Bartonella* are known to be responsible for specific intra-erythrocyte infections in their hosts [144]. *Bartonella* infection is diagnosed by serological, microbiological culture, and/or PCR methods [145]. In West Africa, several species of *Bartonella* have been reported in humans, animals, and arthropods (Figure 6).

*Bartonella henselae* most commonly infects domestic and wild cats but can accidentally infect humans. It is the main causative agent of cat scratch disease, a generally benign disease of children and young adults. In immunocompromised individuals, however, *B. henselae* can cause life-threatening disorders including bacteraemia and bacillary angiomatosis [144]. Further, it is the second most common *Bartonella* species causing endocarditis, with cases having been reported around the world [144]. In West Africa, no cases of *B. henselae* in humans have been reported, but antibodies against the organism have been reported in the serum of cat from Ghana [146]. Around the world, *B. henselae* is mostly transmitted by cat fleas, although the DNA of *B. henselae* has also been found in ticks (*I. pacificus*, *I. ricinus*, and *I. affinis*) [144]. No flea studies have been reported from West Africa, but the DNA of *B. henselae* has been found in *Rh. sanguineus* s.l. collected from dogs, in cats’ blood, and ticks collected from cats in Ghana [73,147].

*Bartonella quintana*, the agent of trench fever, is transmitted by human body lice, although DNA of the organism was found in *Rh. sanguineus* s.l. [148]. The DNA of *B. quintana* has been detected by PCR in blood from febrile patients in rural areas in Senegal [89,149] and antibodies against *B. quintana* were found in the sera of asymptomatic humans who had close contact with fruit bats in Ghana [146].

*Bartonella bovis* is the aetiological agent of bovine bartonellosis, a mostly asymptomatic disease which can occasionally present as endocarditis with anorexia, weight loss, wasting, and abnormal cardiac auscultation [144]. The organism is thought to be transmitted by arthropods such as fleas, flies, lice, mites, and ticks that are found to be naturally infected [150]. Reported in Europe, USA, Asia, and Africa, the prevalence of *B. bovis* in cattle is generally high but varies widely across studies and countries [150]. In West Africa, *B. bovis* has only been reported in cattle from Côte d’Ivoire and Senegal [151,152].

Several *Bartonella* spp. have been found in small mammals in West Africa, including a *Bartonella* spp. closely related *to Bartonella elizabethae*, which was amplified from rodents from Benin [34,153], Nigeria [154], and recently found in Mali [155], a *Bartonella* spp. close to *Bartonella rochalimae* in Benin [153], a *Bartonella* spp. close *to B. tribocorum* in Benin and Nigeria [153,154], a *Bartonella* spp. Similar *to B. grahamii* in Nigeria [154], unidentified *Bartonella* spp. in rodents, *Am. varieagatum*, *Am. latum*, *Hy. rufipes*, *I. aulacodi*, *Rh*. *muhsamae*, *Rh. sulcatus*, and *Rh. microplus* from Benin and Togo [34,36,153], in rodents, bats, and their ectoparasites from Nigeria [154,156], and rodents from Mali [155]. In addition, two potentially new species of *Bartonella* have been described, namely *Candidatus* Bartonella davoustii in cattle in Senegal [151] and *Candidatus* Bartonella mastomydis in rodents in Benin [153] and Mali [155]. Finally, three potentially new genotypes have been identified in rodents from Senegal [62], and *B. senegalensis* and *B. massiliensis*, two newly recognised species, have been reported in *O. sonrai* collected in Senegal [61]. The different *Bartonella* species detected in West Africa in humans, animals, and ticks are shown on the map in Figure 6.

### 2.6. Coxiella burnetii Infection

*Coxiella burnetii* is a small gram-negative intracellular bacterium which is highly resistant to the environment and the causative agent of Q fever. This zoonosis is mainly transmitted to humans by aerosols generated from infected placentas and birth liquids [157]. The organism has been found in many tick species, suggesting these arthropods might also play a role in the transmission of the bacterium [157]. Q fever has been reported almost everywhere it has been investigated, with the exception of New Zealand. Most primary humain infections are asymptomatic, but there can be fever, malaise, headache, fatigue, and pneumonia hepatitis, resulting in acute Q fever [157]. Serious obstetric complications and foetal malformations may occur in pregnant women. About 5% of infections become persistent with endocarditis, lymphadenitis, and vascular and osteoarticular infections developing many years later [157]. In animals, *C. burnetii* infections are also mostly asymptomatic but there might be abortions and stillbirths in cats and domestic ruminants which are a source of infection for people. *Coxiella burnetii* infection can be diagnosed by several methods, including culture (only performed by reference laboratories), serology (the most commonly used method), pathology–immunohistochemistry, and PCR [157].

In West Africa, Q fever has been reported in a man returning from Guinea Bissau with acute lobar pneumonia and fever, headache, haematuria, and hepatitis [158]. It has also been reported in tourists who had stayed in Gambia, Côte d’Ivoire, and Burkina Faso [18]. Antibodies reactive to *C. burnetii* have been reported in Nigerian patients hospitalised for various acute medical conditions [159], apparently healthy adults from Mali, Burkina Faso, and Côte d’Ivoire [77], children aged between one month and five years in Niger [160], children and adults from Ghana [161], nomads in rural areas of northern Burkina Faso [162], in northern Togo [163], blood donors and patients in Mauritania [19], and in adults and children in Gambia [164,165]. *Coxiella burnetii* DNA was detected in febrile patients who tested negative for malaria with Paracheck^®^ in Mali [46] and in human and various environmental samples from Senegal [43,69,89].

In animals from West Africa, antibodies against *C. burnetii* have been detected in sera and milk from cows and in sheep in Nigeria [166,167] as well as in sera from dogs in Senegal and Côte d’Ivoire [168], livestock in the Republic of Guinea [20], cattle, sheep, and goats in Togo, Côte d’Ivoire, Mali, and Ghana [163,169,170,171], small ruminants (goats and sheep) in Gambia [164,172], and rodents in Cape Verde [173]. The DNA of *C. burnetii* has been detected in milk offered for consumption and rodents in Senegal [174,175], rodents in Nigeria [70], and recently in rodents in Mali [155].

The DNA of *C. burnetii* has been found in several species of ticks, although no specific tick vector has been found to play a role in the epizootic cycle of *C. burnetii* and the presence of DNA in a tick does not indicate it plays a role in transmission. In Senegal, DNA of *C. burnetii* was detected in *Am. variegatum*, *Rh.* (*B*.) *annulatus*, *Rh.* (*B*.) *decoloratus*, *Rh. e. evertsi*, *Rh. guilhoni*, *Hy. rufipes*, *Hy. truncatun*, and *O. sonrai* [43] (Table 1). It has also been identified in *Am. variegatum*, *Rh.* (*B*.) *annulatus*, *Rh. e.* evertsi, *Hy. truncatun*, *Hy. dromedarii*, and *Hy. impeltatum* collected from cattle in Nigeria and *Am. variegatum* collected from cattle in Côte d’Ivoire [24,30,49] (Table 1). Recently, *C. burnetii* has been identified in *Am. variegatum*, *Rh.* (*B*.) *microplus*, *Rh. sanguineus* s.l., *Rh. e. evertsi*, *Hy. rufipes*, and *Hy. truncatun* collected from cattle, *Haemaphysalis* spp. from rodents and head lice from patients in Mali [23,155,176], in *Am. variegatum*, *Rh. sanguineus* s.l., *Rhipicephalus* sp., *Hy. truncatun*, and *Hy. rufipes* collected from cattle in Ghana [35], and in unspecified ticks collected from cats in Ghana [147] (Table 1).

## 3. Protozoal Diseases

### 3.1. Theileriosis

Theileriosis is a tropical disease caused by obligate intracellular haemoprotozoa of *Theileria* genus that infect both leukocytes and erythrocytes and are transmitted by hard tick genera, mainly *Amblyomma*, *Haemaphysalis*, *Hyalomma*, and *Rhipicephalus* [177]. Wild and domestic ruminants are the main hosts in which infections cause enlargement of the lymph nodes, fever, anaemia, jaundice, leukopaenia, pulmonary oedema, lethargy, thrombocytopenia, and death [178]. Differential diagnosis of theileriosis requires observation of clinical signs, the epidemiological context, and the use of pathological, microscopic, and molecular techniques to detect the parasite or its DNA [179]. A large number of species have been reported in domestic animals around the world, including *Theileria parva*, *T. annulata*, *T. taurotragi*, *T. lestoquardi*, *T. orientalis*, *T. velifera*, *T. mutans*, *T. uilenbergi*, *T. lowenshuni*, *T. sinensis*, *T. ovis*, and *T. equi* [178]. To date, the species of *Theileria* reported from West Africa are *T. annulata*, *T. mutans*, *T. velifera*, *T. taurotragi*, *T. equi*, and *Theileria* sp. (Figure 7).

*Theileria annulata*, the agent of tropical theileriosis manifested by fever and lymphadenomegaly, is transmitted by several species of *Hyalomma* and is present around the Mediterranean basin, northeast Africa, the Middle East, India, and South Asia. In West Africa, the DNA of *T. annulata* has been detected in cattle from Burkina Faso and Benin [42,44]. In ticks, the DNA of *T. annulata* was found in *Am. variegatum*, *Hy. rufipes*, *Rh. decoloratus*, *Rh. geigyi*, and *Rh. microplus* from Burkina Faso and Benin [42,44], and in *Hy. dromadarii* and *Hy. rufipes* from Mauritania [50] (Table 1).

*Theileria mutans* infects buffalo and cattle in Africa and is transmitted by *Amblyomma* spp. and infections are mostly benign and only very rarely associated with serious clinical illness. In West Africa, *T. mutans* has been detected by microscopy in blood smears from asymptomatic cattle from Senegal [107,108,109], Nigeria [55], and Ghana [96]. The DNA of *T. mutans* has been found in cattle blood from Burkina Faso and Benin [42,44]. In ticks, the DNA of *T. mutans* was detected in *Hy. impeltatum* and *Rh. annulatus* collected from cattle in Nigeria [30] and in *Am. variegatum*, *Hy. rufipes*, *Rh. microplus*, *Rh. geigy*, and *Rh. decoloratus* collected from cattle in Burkina Faso and Benin [39,42,44] (Table 1).

*Theileria velifera* infections in buffalo and cattle in Africa are benign and transmitted by *Amblyomma* spp. [178]. In West Africa, *T. velifera* has been detected in cattle from Nigeria, Ghana, Burkina Faso, and Benin [42,44,55,96]. In ticks, the DNA of *T. velifera* has been detected in *Am. variegatum*, *Hy. rufipes*, *Rh. decoloratus*, *Rh. annulatus*, and *Rh. geigyi* collected from cattle in Burkina, Benin, and Guinea [42,44,45] (Table 1).

*Theileria taurotragi* is a parasite that exists in African eland and cattle and is transmitted by *Rhipicephalus* spp. and infections are benign [178]. In West Africa, *T. taurotragi* has been reported in cattle from Nigeria [55] (Table 1).

*Theileria equi* infects horses, donkeys, and zebras in tropical and subtropical areas, causing equine piroplasmosis which can present as fever, anaemia, inappetence, oedema, icterus, haepatomegaly, splenomegaly, and death in some cases. The disease can cause serious economic losses in the equine industry and is transmitted by several tick genera including *Hyalomma*, *Dermacentor*, *Rhipicephalus*, and *Amblyomma* [178]. In West Africa, antibodies against *T equi* have been detected in horses and donkeys from Nigeria [180,181]. *Theileria equi* DNA has been detected in horses, donkeys, and asymptomatic dogs from Nigeria [38,51,182,183,184], and in horses from Ghana [185] and Senegal [186]. In ticks, *T. equi* DNA was detected in *Rh. e. evertsi*, *Hy. dromedarii*, and *Hy. truncatum* collected from horses and donkeys in Nigeria [51].

In addition, *Theileria* sp. (sable) and a *Theileria* sp. close to *T. ovis* have been identified in Nigerian dogs [38,141]. A *Theileria* sp. has been found in sheep and goats from Ghana [96]. In Senegal, a potentially new species of *Theileria*, provisionally named *Theileria* sp. “Africa”, has been found in a horse [186], and *Candidatus* Theileria senegalensis has been detected in a rodent [62]. The different *Theileria* species detected in West Africa in humans, animals, and ticks are shown on the map in Figure 7.

### 3.2. Babesiosis

Worldwide, there are more than 100 species of *Babesia* which are usually transmitted by hard ticks and infect the erythrocytes of a wide variety of domestic and wild animals. In Africa, animal babesiosis is most commonly due to *B. bigemina* and *B. bovis* in cattle, *B. gibsoni* and *B. canis* in dogs, and *B. caballi* in horses and donkeys [187]. Clinical signs result from haemolysis and include fever, depression, anaemia, splenomegaly, and jaundice. Only a few species infect humans [188], including *Babesia microti*, which mainly infects rodents and is transmitted by *Ixodes* in the northern hemisphere [189], and is the main causative agent of human babesiosis in the United States [189]. *Babesia divergens* is the major cause of human babesiosis in Europe and is transmitted by *Ixodes ricinus*, the only known vector [190]. Human babesiosis caused by *B. microti* and *B. divergens* is characterised by a fever that usually resolves spontaneously in immunocompetent patients. Sometimes there is haematuria and jaundice, depending on the degree of haemolysis [188]. Severe complications of babesiosis, such as splenic infarction and rupture, are more frequently observed in young, immunocompetent male patients [191,192,193]. Babesiosis can be diagnosed by observing abnormalities such as haemolytic anaemia, thrombocytopaenia, elevated transaminases, or by microscopic observation of the parasite on blood smears or detection of the DNA by PCR [17]. In West African humans, a single probable case of human babesiosis has been reported in an infant in Côte d’Ivoire [194]. Several species of *Babesia* have been reported in animals and ticks in West Africa (Figure 8). *Babesia microti* has been found in house rats collected in Nigeria [195] and *B. divergens* DNA has been detected in *Am. variegatum* ticks in Nigeria [28] (Table 1).

*Babesia bigemina* is a parasite of cattle that has been reported on all continents, it is transmitted principally by *Rh.* (*B.*) *microplus* and *Rh.* (*B*.) *annulatus* [196]. In West Africa animals, *B. bigemina* has been detected by microscopy in bovine blood smears in Senegal [107,108,109] and Ghana [96]. The antibodies against *B. bigemina* have been found in cattle in Gambia [197] and Mali [198], and the DNA of *B. bigemina* has been detected in cattle in Nigeria, Burkina Faso, Ghana, Côte d’Ivoire, and Benin [55,112,115]. In ticks, the DNA of *B. bigemina* has been detected in *Rh.* (*B.*) *decoloratus* and *Am. variegatum* from Nigeria [28,64], and *Rh.* (*B*.) *microplus* from Guinea [65] (Table 1).

*Babesia bovis*, the agent of redwater fever in African cattle, is widespread in tropical and subtropical regions around the world and is transmitted by ticks, principally *Rh.* (*B.*) *microplus* and *Rh.* (*B*.) *annulatus*. In West Africa, antibodies against *B. bovis* have been found in cattle in Mali, Nigeria [199,200], and Côte d’Ivoire [201]. Studies have also reported the presence of *B. bovis* DNA in cattle from Nigeria, Burkina Faso, Côte d’Ivoire, Benin, and Ghana [42,44,55,112,115,202].

*Babesia gibsoni* mainly infects dogs in Asia, Europe, America, and North and East Africa, where it is probably transmitted by *Rh. sanguineus* s.l. [187]. *Babesia gibsoni* has only been detected in dogs from Cape Verde in West Africa [203].

There are three subspecies of *Babesia canis*, namely *canis*, *vogeli*, and *rossi*. The first two infect dogs around the world and are transmitted mainly by *Rh. sanguineus* s.l. and *D. reticulatus*. *Babesia canis rossi* infects dogs mainly in Africa, where it is transmitted by *Haemaphysalis leachii* [187] and causes a more severe disease than the two other subspecies. In West Africa, all three subspecies of *B. canis* have been detected in dogs from Nigeria [139,204,205,206].

*Babesia caballi* is also an agent of equine piroplasmosis (see theileriosis above) which occurs in most countries of the world where the competent *Dermacentor*, Rhipicephalus, and *Hyalomma* vectors are found [207]. The DNA of *B. caballi* has been detected in horses from Senegal [186] and Nigeria [183,184] and in *Am. variegatum and Rh. decoloratus* collected from cattle in Guinea [45,46] and *Hyalomma* spp. from Nigeria [49].

*Babesia perroncitoi* and *B. trautmanni* are responsible for swine babesiosis and have been reported in pigs in Nigeria and Ghana [208,209]. The different *Babesia* species detected in West Africa in humans, animals, and ticks are shown on the map in Figure 8.

### 3.3. Hepatozoonosis

Although there are hundreds of *Hepatozoon* species with a very wide variety of hosts and vectors [210], there are no human strains and only two species are of veterinary importance. Both species infect dogs, with *Hepatozoon canis* occurring around the world and *Hepatozoon americanum* infecting dogs in the United States [211]. *Hepatozoon canis* is transmitted by *Rh*. *sanguineus* s.l., with infections resulting in dogs being asymptomatic or developing serious and sometimes fatal signs including fever, lethargy, anaemia, cachexia, weight loss, and hind limb weakness [211]. In West Africa, *H. canis* DNA has been reported in dogs in Nigeria [117,118,140,212], Cape Verde [120,121,203], and Ghana [73,118]. Recently, *H. canis* and two *Hepatozoon* spp., closely related to *Hepatozoon* spp. from snakes in the north of Africa have been detected in rodents from Senegal [62].

In ticks, the DNA of *H. canis* was detected in Nigeria and Ghana [118,140].

## 4. Viral Diseases

### 4.1. Crimean–Congo Haemorrhagic Fever (CCHF)

Crimean-Congo hemorrhagic fever (CCHF) is a zoonotic hemorrhagic disease with a high mortality rate in humans, caused by a virus of the Bunyaviridae family. While CCHF occurs worldwide, it is considered endemic in certain countries in Asia, Europe, and Africa. The virus is transmitted to humans either by ticks (especially of the *Hyalomma* genus) or by contact with the blood or secretions of infected animals [213]. The disease manifests as fever, chills, headache, dizziness, neck pain, nuchal rigidity, photophobia, retro-orbital pain, myalgia, arthralgia, nausea, vomiting, diarrhoea, and abdominal pain. It is associated with coagulopathies manifested by petechia, bruising, haematemesis, and melena, often associated with thrombocytopaenia and leukopaenia [213]. The diagnosis of CCHF is generally made by culture (virus isolation), serology (search for specific IgG and IgM antibodies), or molecular tools (detection of virus RNA) [214].

In West Africa, the first human case of CCHF was reported in 1983 in a febrile patient in southern Mauritania [215]. The CCHF virus has been isolated from a deceased patient, and antibodies against the virus have been found in hospitalised patients with signs of haemorrhagic fever in southwestern Mauritania [216]. Between February and August 2003, 38 people were diagnosed with the CCHF virus in Mauritania, 35 of whom resided in Nouakchott [217], and recently two human cases were diagnosed by ELISA and real-time reverse transcription PCR in Mauritania [218]. Anti-CCHF antibodies have been reported in febrile patients from Nigeria [219,220], a febrile patient with conjunctival jaundice, bleeding gums, and haematemesis from Senegal [221], people working in a slaughterhouse in Ghana [47], and Malian patients who had tested negative for *Plasmodium falciparum* and yellow fever, but who had a history of fever, and haemorrhagic, diarrhoeal, or icteric syndromes [222]. One imported case has been reported, in a woman returning to France from Senegal [223]. The CCHF virus RNA has been detected in humans in Nigeria and Mali [219,224].

In animals, antibodies against the CCHF virus have been detected in cattle from northern Nigeria [225], cattle, rodents, sheep, and goats from Mauritania [53,216,217,226], birds from Senegal [54], and cattle from Mali [227].

In ticks, three different genotypes of the CCHF virus have been identified in *Hy. rufipes*, *Am. variegatum*, *Rh. guilhoni*, and *Rh. e. evertsi* collected from cattle and goats in Senegal [48] (Table 1). The virus has been isolated from the immature stages of *Hy. rufipes* collected from a hornbill in Senegal [48]. It has also been detected in *Hy. rufipes* collected from camels and cattle in Mauritania [53], *Hyalomma* collected from cattle in Mali [228], and recently in *Hy. excavatum* and *Am. variegatum* collected from cattle in Ghana [47] (Table 1). *Hyalomma* spp. ticks are considered to be the only known vectors of the CCHF virus, so the discovery of virus RNA in *Amblyomma* spp. and *Rhipicephalus* spp. requires further investigation before these ticks can be considered as vectors.

### 4.2. African Swine Virus

African swine fever (ASF) is caused by the African swine fever virus (ASFV), which is the only member of the Asfarviridae family. ASF is the only DNA virus transmitted by arthropods, notably soft ticks of *Ornithodoros* genus [229]. In Africa and Europe, *Ornithodoros moubata* and *O. erraticus* are involved in the sylvatic transmission cycle [229]. First discovered in Kenya in 1921, ASF initially affected sub-Saharan African countries. In 2007, it was introduced from East Africa, then spread widely in Europe, and in 2018, the virus was introduced into China via Russia [229]. ASF can manifest itself clinically in domestic pigs in several forms, including perinatal, acute, subacute, or chronic forms. Clinical signs vary according to the form of the disease, and are characterised by high fever (i.e., a body temperature of 41–42 °C), loss of appetite, inactivity, dyspnoea and skin hyperaemia, lethargy, anorexia, inactivity, respiratory distress and severe pulmonary oedema, and sometimes sudden death can also be observed without signs of disease [229]. Diagnosis of ASF is based on culture (virus isolation on porcine macrophages), serology (enzyme-linked immunosorbent assays (ELISA), immunoblotting, and indirect immunostaining techniques), or qPCR (detection of virus DNA) [230].

In West Africa, there is little published data on ASF except in Senegal and Nigeria [231]. ASF was first reported in Senegal in 1957 and the disease is known to be endemic, with over 54 outbreaks reported since 1986 [231], then spread to many West African countries, including Côte d’Ivoire in 1996, Togo in 1998, Ghana in 2000, Burkina Faso in 2003, Niger in 2008, Benin in 2009, Liberia 2010, Gambia 2011, and Mali in 2016 [231,232]. In animals, antibodies against ASF have been detected in pigs in Senegal [233,234], Benin [235], Burkina Faso [235], Ghana [235], and Nigeria [236,237]. The DNA of ASFV was detected in pigs from Benin [235,238], Burkina Faso [235,239,240], Côte d’Ivoire [241], Ghana [235,242], Mali [239], Nigeria [235,243,244,245,246,247,248], Senegal [239], and in Togo [235]. In Nigeria, the DNA of ASFV was found in red river hogs (*Potamochoerus porcus*) [249].

In ticks, the DNA of ASFV has been found only in *O. sonrai* collected from pigsties and warthog burrows in Senegal [63].

## 5. Tick-Borne Diseases in Children and Pregnant Women

The incidence of tick-borne diseases in children has been studied little or not at all in many West African countries. In Senegal, studies have reported that the incidence of tick-borne relapsing fever in children can vary from 0.5% to 4.2% in children under 4 and over 10 years old, respectively [87], and 56% in children under 10 years old [83]. In Togo, TBRF DNA has also been found in children aged between 1 and 14 [92]. No studies on tick-borne diseases have been carried out among pregnant women in West Africa. In Tanzania, an East African country, it has been reported that TBRF during pregnancy can lead to the need for early and effective prevention and management of TBR [250]. However, as in many West African countries, access to health services remains a serious problem, especially for rural populations who are most exposed to tick-borne diseases. In most cases, treatment of tick-borne bacterial diseases is based on antibiotics such as doxycycline, clarithromycin, azithromycin, cefuroxime, and azithromycin [13,17]. Pregnant women and children represent a special population, and treatment of tick-borne diseases must follow specific recommendations. For example, doxycycline is not recommended for the treatment of Lyme disease in pregnant women and children under the age of eight, due to the availability of alternatives such as amoxicillin or cefuroxime. However, in cases of ehrlichiosis and anaplasmosis, doxycycline can be used in pregnant women and children, as the benefit of doxycycline is greater than the risk in these patients [17].

## 6. Conclusions

Tick-borne diseases are generally ignored, neglected and underestimated in West Africa. In febrile autochthonous patients or travellers returning from West Africa, the most commonly reported diseases are African tick-bite fever, tick-borne relapsing fever, Q fever and Crimean–Congo haemorrhagic fever. However, cases of co-infection between different tick-borne micro-organisms have not been described in West Africa as they have been in Europe [251]. Human co-infections can manifest themselves as seropositivity without clinical symptoms or co-disease, i.e., the simultaneous clinical expression of infections by two tick-borne micro-organisms. In most cases, co-infection has no impact on the severity of the disease [251]. However, antibodies against a far broader variety of tick-borne pathogens have been found in healthy blood donors in West Africa and many other tick-borne pathogens can be detected in wild and domestic animals and their ticks. Health professionals working with patients returning from West Africa should be aware they might have been infected with a broad range of tick-borne pathogens. Many tick-borne diseases have non-specific signs, and it is only with careful history taking on travel and possible tick-bites that health workers can be alerted to the possibility of tick-borne diseases in their patients. Diagnosis of tick-borne diseases will be carried out using the simplest techniques, and without doubt the most useful for the future will be molecular biology (qPCR and standard PCR). Ticks are, therefore, of major epidemiological interest for surveillance in areas where there are no specialised diagnostic laboratories testing for infections in humans and animals. Matrix-assisted desorption/ionization time-of-flight mass spectrometry (MALDI-TOF MS), which has recently revolutionised tick identification, could be a technique to facilitate the identification of African ticks, considering the low number of tick morphology identification specialists on this continent.

## Figures and Tables

**Figure 1 pathogens-12-01276-f001:**
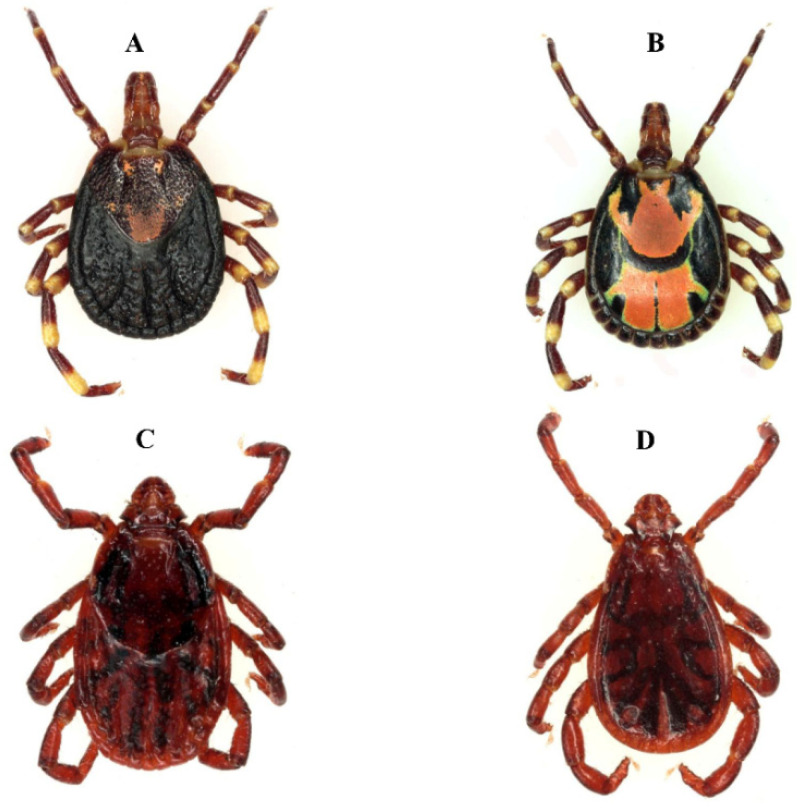
Tick vectors of the spotted fever group rickettsioses in West Africa. Above *Am. variegatum*, vector of *R. africae*, the agent of African tick bite fever, below *Rh. sanguineus*, the main vector of *R. conorii* subsp. *conorii*, the agent of Mediterranean spotted fever. (**A**,**C**) = females, (**B**,**D**) = males.

**Figure 2 pathogens-12-01276-f002:**
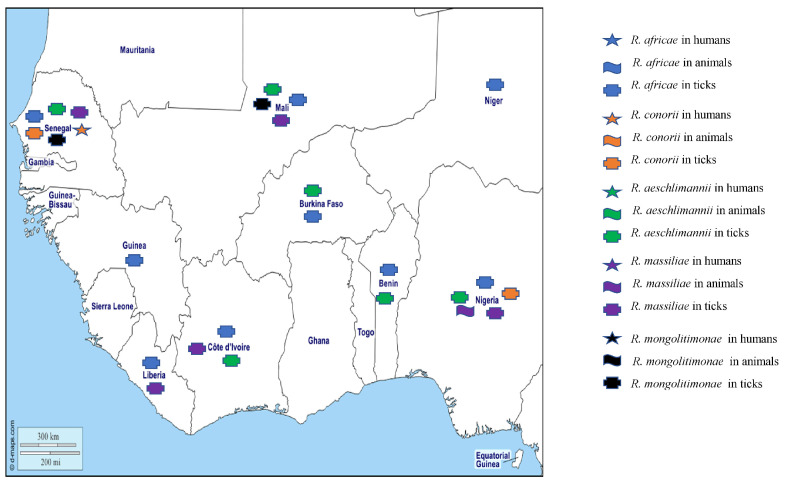
Tick-borne rickettsiae detected by PCR in humans, animals, and ticks in West African countries.

**Figure 3 pathogens-12-01276-f003:**
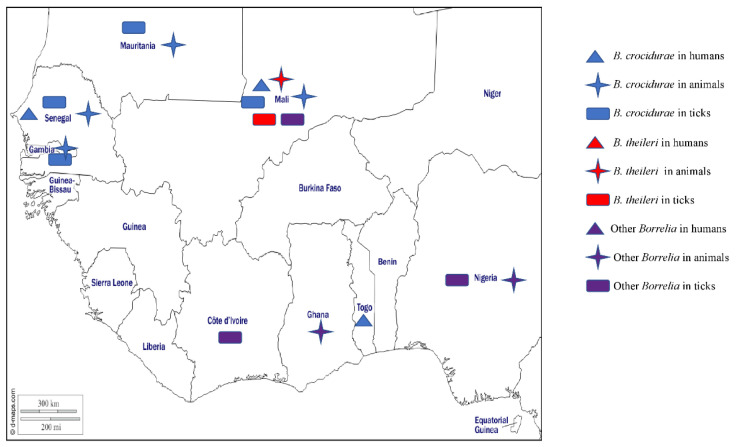
Different species of *Borrelia* detected by PCR or microscopy methods in humans, animals, and ticks in West Africa.

**Figure 4 pathogens-12-01276-f004:**
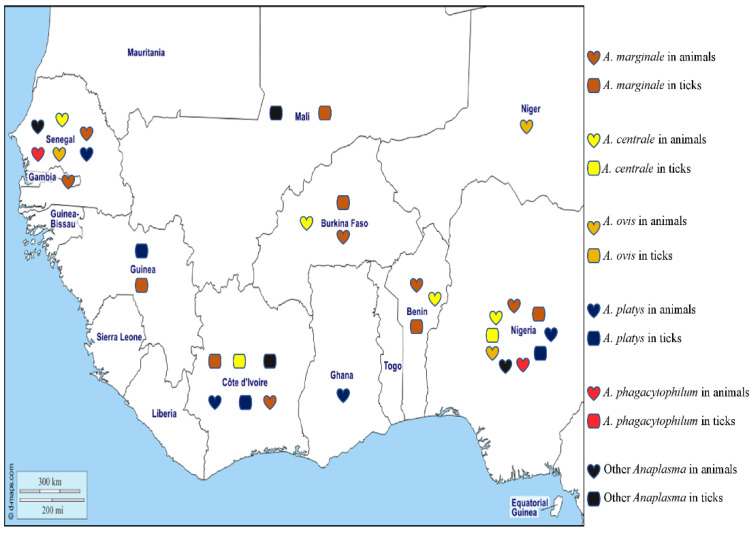
Different species of *Anaplasma* detected by PCR in animals and ticks in West Africa.

**Figure 5 pathogens-12-01276-f005:**
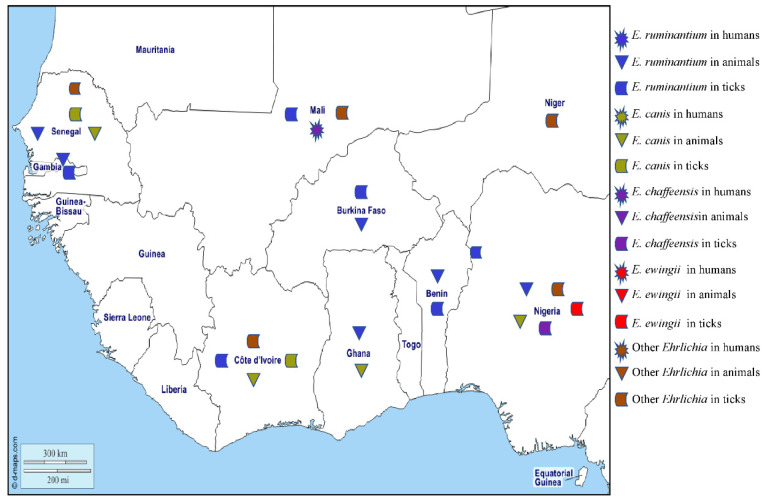
Different species of *Ehrlichia* spp. detected by PCR in humans, animals, and ticks in West Africa.

**Figure 6 pathogens-12-01276-f006:**
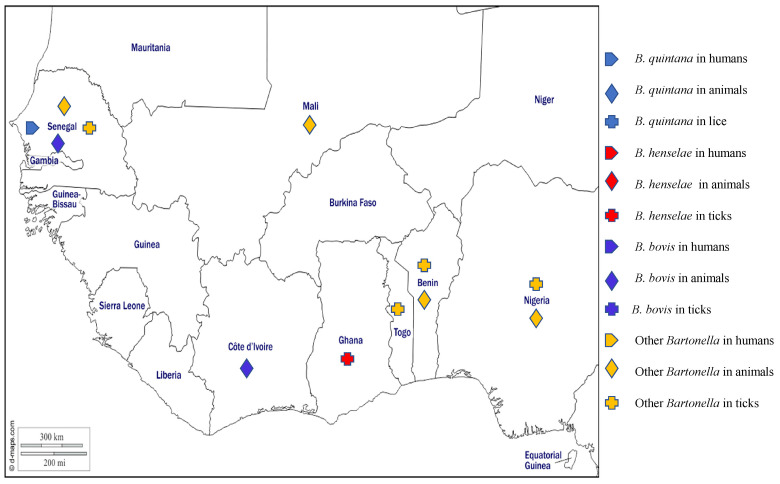
Different species of *Bartonella* spp. detected by PCR in humans, animals, and ectoparasites in West Africa.

**Figure 7 pathogens-12-01276-f007:**
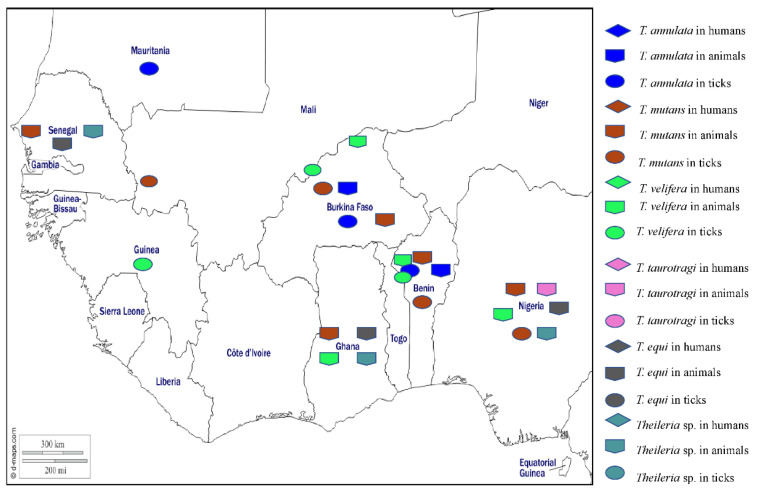
Different species of *Theileria* detected by PCR in humans, animals, and ticks in West Africa.

**Figure 8 pathogens-12-01276-f008:**
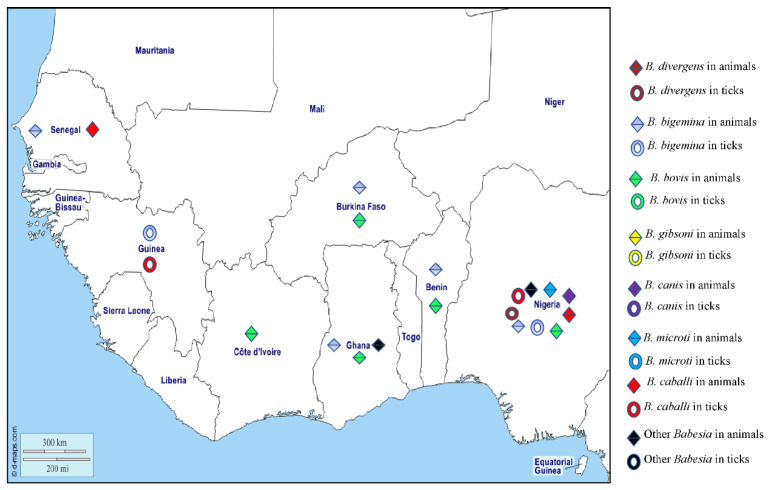
Different species of *Babasia* spp. detected by PCR or microscopy in animals and ticks in West Africa.

**Table 1 pathogens-12-01276-t001:** Tick species found to be positive for tick-associated pathogens in different West African countries.

Tick Species	Organism Detected		Countries	References
			Côte d’Ivoire, Niger, Mali, Liberia, Guinea, Senegal, Benin, Burkina Faso, Nigeria	[22,23,24,25,26,27,28,29,30,31,32,33]
	*R. africae **			
	*R. aeschlimannii*		Benin	[34]
	*Rickettsia* spp.		Benin, Togo, Ghana	[34,35,36]
	*Candidatus* africana	Borrelia	Côte d’Ivoire	[24]
	*Candidatus* ivorensis	Borrelia	Côte d’Ivoire	[24]
	*Borrelia* spp.		Mali	[23]
	*A. marginale*		Benin	[37]
	*A. centrale*		Côte d’Ivoire	[24]
	*Candidatus* ivorensis	Anaplasma	Côte d’Ivoire	[24]
*Am. variegatum*	*E. ruminantium **		Mali, Burkina Benin, Nigeria Côte Faso, d’Ivoire, Gambia,	[23,24,38,39,40,41,42]
	*Ehrlichia ewingii*		Nigeria	[30]
	*Candidatus* Ehrlichia rustica		Côte d’Ivoire	[24]
	*Candidatus* Ehrlichia urmitei		Côte d’Ivoire	[24]
	*Bartonella* spp.		Benin, Togo	[34,36]
	*C. burnetii*		Mali, Côte d’Ivoire, Nigeria, Senegal, Ghana	[23,24,30,43]
	*T. annulata*		Burkina Faso, Benin	[42,44]
	*T. mutans **		Burkina Faso, Benin	[39,42,44]
	*T. velifera **		Guinea, Burkina Faso, Benin	[42,44,45]
	*B. bigemina*		Nigeria	[28]
	*B. divergens*		Nigeria	[28]
	*B. caballi*		Guinea	[46]
	CCHF virus		Ghana, Senegal	[47,48]
*Am. compressum*	*R. africae*		Guinea	[26]
*Am. latum*	*Rickettsia* spp.		Benin	[36]
	*Bartonella* spp.		Benin	[36]
*Hae. paraleachi*	*R. africae*		Guinea	[26]
*Hy. dromedarii*	*R. aeschlimannii*		Nigeria	[49]
	*T. annulate **		Mauritania	[50]
	*T. equi **		Nigeria	[51]
*Hy. excavatum*	CCHF virus		Ghana	[47]
	*R. africae*		Nigeria	[28]
	*R. aeschlimannii*		Senegal, Nigeria	[31,49,52]
*Hy. impeltatum*	*A. centrale*		Nigeria	[30]
	*E. chaffeensis*		Nigeria	[30]
	*E. ewingii*		Nigeria	[30]
	*C. burnetii*		Nigeria	[30]
	*T. mutans **		Nigeria	[30]
*Hy. impressum*	*R. africae*		Côte d’Ivoire	[24]
*Hy. rufipes*	*R. africae*		Mali, Côte d’Ivoire, Guinea, Senegal	[23,24,26,31]
	*R. aesclimannii*		Mali, Senegal, Burkina Faso, Nigeria, Benin	[21,23,31,33,34,49,52]
	*Rickettsia* spp.		Benin, Ghana	[34]
	*Bartonella* spp.		Benin	[36]
	*E. ruminantium*		Mali	[23]
	*C. burnetii*		Mali	[23]
	*T. annulata* *		Burkina Faso, Benin, Mauritania	[42,44,50]
	*T. mutans **		Burkina Faso, Benin	[42,44]
	*T. velifera*		Burkina Faso, Benin	[42,44,45]
	CCHF virus *		Mauritania, Senegal	[48,53,54]
*Hy. truncatum*	*R. africae*		Mali, Côte d’Ivoire	[23,24]
	*R. aeschlimannii*		Mali, Côte d’Ivoire, Senegal, Burkina Faso, Nigeria	[21,23,24]
	*R. mongolitimonae*		Mali, Senegal	[21,23]
	*Rickettsia* spp.		Togo, Ghana	[34]
	*A. platys*		Nigeria	[55]
	*E. ruminantium*		Mali	[23]
	*Ehrlichia* spp.		Niger	[29]
	*Candidatus* Ehrlichia rustica		Mali, Côte d’Ivoire	[23,24]
	*Candidatus* Ehrlichia urmitei		Côte d’Ivoire	[24]
	*C. burnetii*		Mali, Senegal	[23,43]
	*T. equi **		Nigeria	[51]
	CCHF virus *		Mauritania	[48,53,54]
*Ix. aulacodi*	*Rickettsia* spp.		Benin	[36]
	*Bartonella* spp.		Benin	[36]
*O. sonrai*	*B. crocidurae **		Mali, Senegal, Mauritania, Gambia	[56,57,58,59,60]
	*B. senegalensis*		Senegal	[61]
	*B. massiliensis*		Senegal	[61]
	*C. burnetii*		Senegal	[43]
	*R. africae*		Senegal, Nigeria, Guinea	[21,26,30]
	*Ehrlichia* sp.		Senegal	[58,62]
	ASF virus		Senegal	[63]
*Rh.* (*B*.) *annulatus*	*R. africae*		Guinea, Nigeria	[26,28]
	*Rickettsia* spp.		Togo	[34]
	*A. centrale*		Nigeria	[30]
	*E. ewingii*		Nigeria	[30]
	*C. burnetii*		Nigeria, Senegal	[30,43]
	*T. mutans*		Nigeria	[30]
	*T. velifera*		Guinea	[45]
*Rh.* (*B*.) *decoloratus*	*A. marginale **		Nigeria, Burkina Faso	[28,44]
	*C. burnetii*		Senegal	[43]
	*T. annulata*		Burkina Faso, Benin	[42,44]
	*T. mutans*		Burkina Faso, Benin	[42,44]
	*B. bigemina **		Nigeria	[64]
	*T. velifera*		Guinea, Burkina Faso	[44,45]
*Rh.* (*B*.) *geigyi*	*R. africae*		Liberia	[26]
	*A. marginale*		Guinea	[65]
	*T. velifera*		Guinea, Burkina Faso	[44,45]
	*B. theileri*		Mali	[66]
*Rh.* (*B*.) *microplus*	*R. africae*		Mali, Côte d’Ivoire	[23,24]
	*A. marginale*		Mali, Côte d’Ivoire, Benin, Guinea	[23,24,37]
	*T. annulata*		Burkina Faso, Benin	[42,44]
	*T. mutans*		Burkina Faso, Benin	[42,44]
	*A. platys*		Guinea	[65]
	*Candidatus Anaplasma ivorensis*		Mali	[23]
	*E. ruminantium*		Mali, Côte d’Ivoire, Burkina Faso	[23,24,67]
	*Candidatus Ehrlichia urmitei*		Mali, Côte d’Ivoire	[23,24]
	*Candidatus* Ehrlichia rustica		Côte d’Ivoire	[24]
	*C. burnetii*		Mali	[23]
	*T. mutans*		Benin	[37]
	*Rickettsia* spp.		Benin, Togo	[34]
	*Bartonella* spp.		Benin, Togo	[34]
*Rh. e. evertsi*	*R. africae*		Mali, Senegal, Nigeria	[21,26,30]
	*R. aeschlimannii*		Senegal	[21,23,31,33,52]
	*R. conorii conorii*		Senegal	[21]
	*E. ruminantium*		Mali	[23]
	*E. canis*		Senegal	[68]
	*E. chaffeensis*		Nigeria	[30]
	*C. burnetii*		Mali, Nigeria, Senegal	[23,30,69]
	*T. annulata*		Burkina Faso, Benin	[42,44]
	*T. mutans*		Burkina Faso, Benin	[37,44]
	*B. bigemina **		Guinea	[65]
	*T. equi **		Nigeria	[51]
	CCHF virus		Senegal	[54]
*Rh. guilhoni*	*R. massiliae*		Senegal	[21]
	*C. burnetii*		Senegal	[43]
	CCHF virus		Senegal	[54]
*Rh. muhsamae*	*Ehrlichia* sp.		Mali	[29]
	*R. africae*		Nigeria	[28]
	*R. aeschlimannii*		Mali	[23]
	*R. massiliae*		Nigeria	[27]
	*Bartonella* spp.		Togo	[34]
*Rh. sanguineus* s.l.	*R. conorii conorii **		Nigeria	[70]
	*Rickettsia* spp.		Benin, Ghana	[36]
	*A. platys*		Côte d’Ivoire	[71]
	*E. canis **		Côte d’Ivoire	[72]
	*Candidatus Neoehrlichia mikurensis*		Nigeria	[55]
	*B. henselae*		Ghana	[73]
	*C. burnetii*		Mali	[23]
*Rh. senegalensis*	*R. massiliae*		Côte d’Ivoire, Liberia	[24,26]
*Rh*. *sulcatus*	*Bartonella* spp.		Togo	[34]
*Rh. turanicus*	*R. massiliae*		Nigeria	[74]

* established as transmitted by the tick species.

## Data Availability

Not applicable.

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
