# Peer review of "Tick-Borne Diseases of Humans and Animals in West Africa"

_pathogens, 2023, doi:10.3390/pathogens12111276_

Round 1

Reviewer 1 Report

This is a comprehensive review on the topic. In order to improve the paper I would suggest the following:

1. Please add how climate change contributes to the increase in incidence of tick-borne diseases, and particularly how it is affecting West Africa compared to, for example Europe or North America

2. Section regarding Anaplasmosis should be updated with newer references, ref 71 seems to be obsolete. Additionally, the presence of thrombocytopenia and alteration in liver enzymes is the most prominent laboratory feature and should be added. 

3. Section 3.2 , Babesiosis- it should be noted that severe complication of babesiosis, splenic infarct and rupture are more common in immunocompetent young male patients ( Splenic Complications of Babesia microti Infection in Humans: A Systematic Review - PubMed (nih.gov) , Splenic Rupture as the First Manifestation of Babesia Microti Infection: Report of a Case and Review of Literature - PubMed (nih.gov), Babesiosis-associated Splenic Rupture: Case Series From a Hyperendemic Region | Clinical Infectious Diseases | Oxford Academic (oup.com)

4. Authors should consider adding few lines in each section about diagnostic approach of tick borne diseases ( Tickborne Diseases: Diagnosis and Management - PubMed (nih.gov))

5. In discussion I would like to see more information regarding co-infection between different pathogens and how it affects clinical picture

6. One paragraph should be dedicated to special population- children and pregnant females and how treatment of these pathogens differs in this population

minor editing and grammatical changes needed

Author Response

Reviewer 1

Comments and Suggestions for Authors

This is a comprehensive review on the topic. In order to improve the paper I would suggest the following:

  1. Please add how climate change contributes to the increase in incidence of tick-borne diseases, and particularly how it is affecting West Africa compared to, for example Europe or North America

Authors' response: We thank the reviewer, we have added the following paragraph in introduction part “Tick-borne diseases are strongly influenced by many factors, including host distribution, tick abundance and seasonality, pathogen virulence, climate (temperature, precipitation, humidity and vegetation cover) all of which contribute to the emergence and re-emergence of tick-borne diseases. Climate change may impact the incidence of tick-borne diseases by increasing tick populations, the rate of contact between livestock and ticks, and the rate of contact between livestock and wildlife [8]. In Europe, studies have been carried out on the impact of climate change on tick-borne diseases. It has been reported that climate change is responsible for the extension of the range of Ixodes ricinus in the north and at higher altitudes, leading to an increase in the prevalence of tick-borne en-cephalitis. Climate change is also partly responsible for changes in the distribution of Dermacentor reticulates [9]. Over the last 20 years, the incidence of tick-borne diseases (Lyme disease, tick-borne encephalitis and Crimean-Congo haemorrhagic fever) has in-creased in both Europe and America [10]. Although this increase could be partly caused by climate change, other factors could also contribute, as tick-borne disease systems are quite complex [10]. In Africa, diseases transmitted by ticks to humans are poorly studied, making it difficult to measure the impact of climate change on these diseases. However, there has been an increase in the incidence of tick-borne relapsing fever, one of the best-studied tick-borne bacterial diseases in Senegal since the 1970s, and its range has extend-ed over 350 km to north-west Morocco, due to increased drought conditions [10].”

  1. Section regarding Anaplasmosis should be updated with newer references, ref 71 seems to be obsolete. Additionally, the presence of thrombocytopenia and alteration in liver enzymes is the most prominent laboratory feature and should be added. 

Authors' response: We thank the reviewer, we have updated the most recent references and the reference 71 has been deleted. Thrombocytopenia and an alteration of liver enzymes is the most striking laboratory feature have been added as follows “In humans, A. phagocytophilum can be found in circulating neutrophils and is the agent of human granulocytic anaplasmosis (HGA), manifested by lethargy, inappetence, weight loss, musculoskeletal pain respiratory insufficiency and severe gastrointestinal bleeding [99,100]. Thrombocytopenia and liver enzyme alterations are the most common laboratory abnormality in HGA [99].”

  1. Section 3.2 , Babesiosis- it should be noted that severe complication of babesiosis, splenic infarct and rupture are more common in immunocompetent young male patients ( Splenic Complications of Babesia microti Infection in Humans: A Systematic Review - PubMed (nih.gov) , Splenic Rupture as the First Manifestation of Babesia Microti Infection: Report of a Case and Review of Literature - PubMed (nih.gov), Babesiosis-associated Splenic Rupture: Case Series From a Hyperendemic Region | Clinical Infectious Diseases | Oxford Academic (oup.com)

Authors' response: We thank the reviewer, we have updated this section with the information you provided as follows “Severe complications of babesiosis, such as splenic infarction and rupture, are more frequently observed in young, immunocompetent male patients [198-200].”

  1. Authors should consider adding few lines in each section about diagnostic approach of tick borne diseases ( Tickborne Diseases: Diagnosis and Management - PubMed (nih.gov))

Authors' response: We thank the reviewer, as requested by the reviewer, diagnostic techniques for each of these tick-borne diseases have been added to the various paragraphs.

  1. In discussion I would like to see more information regarding co-infection between different pathogens and how it affects clinical picture

Authors' response: We thank the reviewer, as requested by the reviewer, we have added in conclusion information regarding co-infection between different pathogens and how it affects clinical picture as follows “However, cases of co-infection between different tick-borne micro-organisms have not been described in West Africa as they have in Europe [258]. Human co-infections can manifest themselves as seropositivity without clinical symptoms or co-disease, i.e. the simultaneous clinical expression of infections by two tick-borne micro-organisms. In most cases, co-infection has no impact on the severity of the disease [258].”

  1. One paragraph should be dedicated to special population- children and pregnant females and how treatment of these pathogens differs in this population

Authors' response: We thank the reviewer, as requested by the reviewer one paragraph have been added to special population- children and pregnant females as follows

 “5. Tick-borne diseases in children and pregnant women

The incidence of tick-borne diseases in children has been studied little or not at all in many West African countries. In Senegal, studies have reported that the incidence of tick-borne relapsing fever in children can vary from 0.5% to 4.2% in children under four and over 10, respectively [87], and 56% in children under 10 [83]. In Togo, TBRF DNA has al-so been found in children aged between one and 14 [92]. No studies on tick-borne diseas-es have been carried out among pregnant women in West Africa. In Tanzania, an East Af-rican country, it has been reported that TBRF during pregnancy can lead to the need for early and effective prevention and management of TBR [257]. However, as in many West African countries, access to health services remains a serious problem, especially for ru-ral populations. In most cases, treatment of tick-borne bacterial diseases is based on anti-biotics such as doxycycline, clarithromycin, azithromycin, cefuroxime and azithromycin [13,17]. Pregnant women and children represent a special population, and treatment of tick-borne diseases must follow specific recommendations. For example, doxycycline is not recommended for the treatment of Lyme disease in pregnant women and children under the age of eight, due to the availability of alternatives such as amoxicillin or ce-furoxime. However, in cases of ehrlichiosis and anaplasmosis, doxycycline can be used in pregnant women and children, as the benefit of doxycycline is greater than the risk in these patients [17].”

Reviewer 2 Report

"Tick-Borne Diseases of People and Animals in West Africa" is a well-written and easy-to-read review article. It provides precise information about the presence and distribution of tick-borne diseases in humans, animals, and ticks.

Author Response

Authors' response: We thank the reviewer

Reviewer 3 Report

The manuscript "Tick-Borne Diseases of People and Animals in West Africa" is written in the review form and brings valuable contributions to the field of tick-borne diseases.  The authors overview current knowledge of ticks and tick-borne pathogens' presence and distribution in West Africa in an appropriate and sufficient manner.  I would only suggest changing the title to "Tick-Borne Diseases of Humans and Animals in West Africa". 

Author Response

Reviewer 3

Comments and Suggestions for Authors

The manuscript "Tick-Borne Diseases of People and Animals in West Africa" is written in the review form and brings valuable contributions to the field of tick-borne diseases.  The authors overview current knowledge of ticks and tick-borne pathogens' presence and distribution in West Africa in an appropriate and sufficient manner.  I would only suggest changing the title to "Tick-Borne Diseases of Humans and Animals in West Africa". 

Authors' response: We thank the reviewer, we have changed the title as requested by the reviewer as follows “Tick-Borne diseases of humans and animals in West Africa”

Round 2

Reviewer 1 Report

I would like to thank the authors on detailed revisions of the paper. I think this version is significantly improved from pervious and is acceptable for publication in the current form. The authors have addressed all shortcomings and I expect this paper to be well cited since it provides novel data on very important and emerging topic. Congratulations!

accept as such